# TOWARDS REVERSIBLE MODEL MERGING FOR LOW-RANK WEIGHTS

## ABSTRACT

Model merging aims to combine multiple fine-tuned models into a single set of weights that performs well across all source tasks. While prior work has shown that merging can approximate the performance of individual fine-tuned models for each task, it largely overlooks scenarios where models are compressed into low-rank representations—either through low-rank adaptation (LoRA) or post-training singular value decomposition (SVD). We first demonstrate that applying conventional merging methods to low-rank weights leads to severe performance degradation in the merged model. Motivated by this phenomenon, we propose a fundamentally different approach: instead of collapsing all adapters into one set of weights, we construct a compact basis (e.g., an equivalent of holding two or more models) from which original task-specific models can be recovered via linear combination. This reframes merging as generating a reconstruction-capable model space rather than producing a single merged model. Crucially, this allows us to "revert" to each individual model when needed, recognizing that no merged model can consistently outperform one specialized for its task. Building on this insight, we introduce our method, Reversible Model Merging (RMM), an efficient, data-free, and flexible method that provides a closed-form solution for selecting the optimal basis of model weights and task-specific coefficients for linear combination. Extensive experiments across diverse datasets and model scales demonstrate that RMM consistently outperforms existing merging approaches, preserving the performance of low-rank compressed models by a significant margin.

## 1 INTRODUCTION

Parameter-efficient fine-tuning methods, such as Low-Rank Adapters (LoRA) (Hu et al., 2021), have become a cornerstone for adapting large language models to diverse downstream tasks without incurring the full cost of retraining. In parallel, model compression techniques such as post-training singular value decomposition (SVD) truncation have been widely adopted to reduce storage and inference costs, offering an additional dimension of efficiency (Ryu et al., 2023; Ping et al., 2024; Wang et al., 2025b;a). Beyond their conceptual simplicity, post-training SVD compression methods are also used in practice because they require no access to training data and impose zero additional training cost, making them especially attractive in deployment-time or model-serving scenarios where fine-tuning is not possible. Complementing these approaches, the intrinsic-dimensionality perspective of Aghajanyan et al. (2021) demonstrates that downstream tasks often lie in extremely low-dimensional subspaces—sometimes requiring only a few hundred effective parameters—providing theoretical and empirical support for why low-rank adaptation and SVD-compression-based methods are so effective in practice. Alongside these advances, model merging has emerged as a promising paradigm for consolidating the knowledge of multiple task-specific models into a single unified model. Recent advances in merging, ranging from simple parameter interpolation to optimization-based techniques, have demonstrated that merged models can perform competitively across source tasks, without requiring access to the original training data (Matena & Raffel, 2022; Yang et al., 2024a; Yadav et al., 2023; Wei et al., 2025).

Despite this progress, even the most sophisticated merging strategies consistently fail to match the performance of the original individual fine-tuned models. This performance gap is particularly pronounced when merging models that have been compressed into low-rank form, whether via LoRA fine-tuning or post-training SVD truncation. While merging enables a single model to handle multi-

ple tasks, it inherently collapses distinct task-specific representations into a single-parameter space. This often leads to interference between tasks and erodes the specialized adaptations learned by each model (Gargiulo et al., 2025).

Although low-rank adapters and SVD-truncated models are inherently compact, the proliferation of such compressed models introduces a new scalability bottleneck. In practical multi-task, federated or continual learning scenarios, it is common to accumulate dozens or even hundreds of task-specific low-rank adapters, each requiring separate storage, and management coordination. Retaining all of them diminishes the storage savings that low-rank parameterization was designed to provide, and makes deployment more cumbersome. This motivates the need for merging strategies that operate directly on low-rank representations—not only to consolidate storage and reduce redundancy, but also to enable more efficient sharing, deployment, and transfer of compressed models across tasks.

To better understand the limitations of current approaches, we first investigate the behavior of existing merging techniques when applied to low-rank compressed models. Specifically, we focus on deltas, i.e., the weight differences between each fine-tuned model and the shared pre-trained initialization. Conventional merging strategies attempt to combine these deltas into a single set of parameters, assuming linear compatibility. However, we empirically show that merging low-rank deltas directly leads to catastrophic performance degradation. This failure stems from two key factors: the limited expressive capacity inherent in low-rank representations, and misalignment of task-specific subspaces, each optimized independently, leading to severe interference when combined. The intuition is that task interference already arises even when merging full, uncompressed models (Gargiulo et al., 2025). When each task is further constrained to a highly compressed low-rank subspace, this interference intensifies: the representational space available to each task becomes more compact, reducing the degrees of freedom needed to remain robust under merging. As a result, misaligned task-specific subspaces collide more severely, amplifying performance degradation.

Followed by this observation, we take a step back and ask: *What if the goal of merging were not to produce a single final model? What if we could relax the problem to allow retaining more than one model? What if we could preserve the ability to reconstruct the original models on demand?* Instead of collapsing multiple low-rank adapters into a single set of weights, we propose to maintain a compact set of weights that serves as a shared basis from which each original model can be recovered through linear combination. This reframing opens an underexplored design space: rather than compressing all information into a single model, we preserve complementary representational directions across a small set of basis models, enabling post-merging "reversion" to the original adapters. Our approach challenges the prevailing assumption that merging must terminate in a single model, and instead redefines merging as the construction of a reconstruction-capable model space.

Building on this reconstruction-driven perspective, we develop a theoretical framework that characterizes the optimal choice of basis for minimizing reconstruction error. This leads to our method, Reversible Model Merging (RMM), which reinterprets the merging process through a basis-selection framework. We formalize merging as a basis selection problem, where the objective is to identify a compact set of task vectors that enables accurate reconstruction of all original models. Within this framework, RMM provides a closed-form solution that simultaneously identifies an optimal set of basis task vectors and determines the task-specific coefficients for linear combination to recover individual task adaptations. We empirically validate RMM across a broad range of datasets and model configurations, showing that it consistently surpasses conventional merging strategies by a significant margin. RMM achieves this while preserving the efficiency benefits of low-rank parameterization, underscoring its practicality for large-scale multi-task adaptation. Moreover, it offers a flexible trade-off between storage and performance via a tunable hyperparameter that controls the number of final models retained.

## 2 RELATED WORK

Model merging has emerged as a powerful strategy for consolidating knowledge from multiple fine-tuned models. Existing methods broadly fall into two categories: parameter-space manipulations that require no data, and optimization-based approaches that leverage data or auxiliary objectives. Below, we review representative techniques from both lines of work:

**Task Arithmetic**: A foundational idea in model merging is task arithmetic (Ilharco et al., 2023). In this approach, a delta is defined as the difference between the parameters of a fine-tuned model and its pre-trained initialization. The key insight of task arithmetic is that such deltas behave like modular building blocks. They can be added to a base model to impart new abilities, subtracted to suppress unwanted behaviors, or combined through scaling to blend multiple capabilities. This intuitive linear framework established a strong foundation for composing model behaviors directly in weight space.

**DARE**: While simple delta addition can be effective, it often fails when merged models contain conflicting or overlapping parameter updates. To address this, Yu et al. (2024) introduced DARE (Drop and Rescale), a technique built on aggressive randomized pruning. The method operates by discarding the vast majority of parameter changes—up to 90-99%—on the premise that most updates are not essential and may even interfere with knowledge from other models. The remaining updates are then amplified to preserve the overall scale of the modification, preserving core task-specific modifications while minimizing interference.

**TIES**: In contrast to DARE's randomized pruning, TIES-merging introduces a deterministic framework for mitigating parameter conflicts (Yadav et al., 2023). It operates in three phases—Trim, Elect Sign, and Merge—each designed to systematically resolve interference across models. First, only the parameters with the largest magnitudes are retained, ensuring that the most influential updates from fine-tuning remain. Next, for each parameter position, the method evaluates the sign of updates across models and enforces agreement by discarding contributions that oppose the majority sign. Finally, the sign-consistent parameters are averaged to form the merged representation. This consensus-driven approach reduces destructive interference and yields a more stable merged model. Building on the same motivation of reducing interference as in DARE and TIES, EMR-Merging (Elect, Mask and Rescale) proposes electing a unified model and attaching task-specific binary masks and rescalers from each individual model to shift the unified model towards each task (Huang et al., 2024).

**Other Merging Methods**: Beyond purely parameter-space manipulations, several model merging approaches rely on optimization procedures or direct access to some small dataset. Methods such as model soups (Wortsman et al., 2022) refine merging weights for task-arithmetic by minimizing loss on validation data, effectively steering the interpolation toward better generalization. Similarly, Fisher-weighted averaging (Matena & Raffel, 2022) incorporates second-order information to guide the merge, assigning higher importance to parameters that are estimated as more critical for task performance. Other lines of work explicitly formulate merging as an optimization problem: for example, RegMean (Jin et al., 2023) and AdaMerging (Yang et al., 2024a) adaptively search for task-dependent merging coefficients by solving small-scale training objectives on the discrepancy between predictions of the merged and individual models and the entropy of predictions for unlabeled data, respectively. A recent approach formulates model merging as minimizing the loss difference between the merged model and the individual fine-tuned models, using a first-order Taylor expansion to derive a tractable surrogate objective (Wei et al., 2025). While these approaches can yield stronger task-specific performance, they often require access to labeled or unlabeled data, additional optimization, or both, which limits their applicability in settings where merging must be performed without retraining and data.

## 3 PRELIMINARIES

We consider the problem of consolidating a collection of fine-tuned models into a single unified model via model merging. Let $\{\theta_1, \ldots, \theta_n\}$ denote $n$ models fine-tuned on $n$ distinct tasks. A merging algorithm $\mathcal{M}$ combines these models into a single set of weights $\theta_{\mathcal{M}} = \mathcal{M}(\{\theta_i\}_{i=1}^n)$. The objective is to identify a merging procedure $\mathcal{M}$ that maximizes the average validation performance across all tasks:

$$\max_{\mathcal{M}} \frac{1}{n} \sum_{i=1}^{n} \mathrm{Perf}(\theta_{\mathcal{M}}, \mathcal{D}_i), \tag{1}$$

where $\mathcal{D}_i$ denotes a held-out validation set for task $i$, and $\mathrm{Perf}$ denotes any suitable performance metric (e.g., accuracy or F1 score). It is standard to assume that all fine-tuned models originate from the same pretrained initialization $\theta_{\mathrm{pre}}$. For any layer $l$, the update (or delta) of model $i$ relative to the

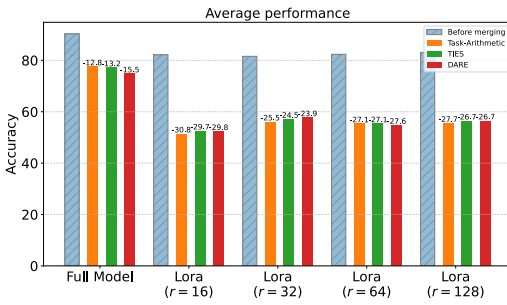 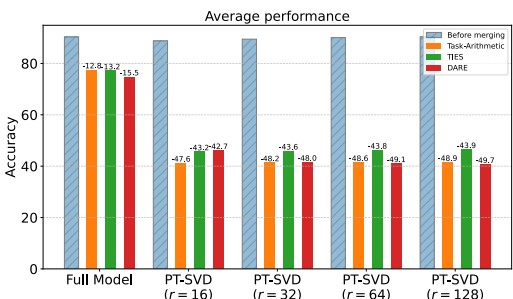

Figure 1: Average task performance of individual models (before merging) versus merged models for different merging methods. Results are shown for fully fine-tuned models and models fine-tuned using LoRA with various ranks $r \in \{16, 32, 64, 128\}$.

Figure 2: Average task performance of individual models (before merging) versus merged models for different merging methods. Results are shown for fully fine-tuned models and models compressed using PT-SVD with various ranks $r \in \{16, 32, 64, 128\}$.

base model is defined as $\boldsymbol{\Delta}_i^l = \boldsymbol{\theta}_i^l - \boldsymbol{\theta}_{\text{pre}}^l$, where $\boldsymbol{\Delta}_i^l \in \mathbb{R}^{m \times d}$ denotes the weight difference for a layer of dimensions $m \times d$.

## 3.1 LOW-RANK COMPRESSED MODELS

In practice, it is common to compress fine-tuned models into low-rank form to reduce storage, training and inference costs. Two widely adopted compression strategies are:

**LoRA fine-tuning**: During training, each layer is augmented with trainable low-rank matrices $\boldsymbol{A} \in \mathbb{R}^{m \times r}$ and $\boldsymbol{B} \in \mathbb{R}^{r \times d}$, where $r \ll \min(m, d)$. The pretrained weights remain frozen, and only $\boldsymbol{A}$ and $\boldsymbol{B}$ are optimized.

**Post-training SVD truncation (PT-SVD)**: After standard fine-tuning, the update $\boldsymbol{\Delta}_i^l$ is decomposed via SVD and truncated to retain only the top-$r$ singular components, yielding a low-rank approximation $\hat{\boldsymbol{\Delta}}_i^l = \boldsymbol{A}_i^l \boldsymbol{B}_i^l$, for $\boldsymbol{A}_i^l \in \mathbb{R}^{m \times r}$ and $\boldsymbol{B}_i^l \in \mathbb{R}^{r \times d}$ where $r \ll \min(m, d)$.

In both cases, the fine-tuned update for layer $l$ in model $i$ is represented in low-rank form $\hat{\boldsymbol{\Delta}}_i^l = \boldsymbol{A}_i^l \boldsymbol{B}_i^l$, where $\boldsymbol{A}_i^l \in \mathbb{R}^{m \times r}$ and $\boldsymbol{B}_i^l \in \mathbb{R}^{r \times d}$, with $r$ denoting the rank of compression. Thus, the merging problem becomes one of combining low-rank updates across tasks. The goal is to construct a merged model that integrates the compressed deltas $\hat{\boldsymbol{\Delta}}_i = \{\hat{\boldsymbol{\Delta}}_i^l\}_{l=1}^L$ across all the tasks:

$$\hat{\theta}_{\mathcal{M}} = \theta_{\text{pre}} + \mathcal{M}(\{\hat{\boldsymbol{\Delta}}_i\}_{i=1}^n), \quad \max_{\mathcal{M}} \frac{1}{n} \sum_{i=1}^n \text{Perf}(\hat{\theta}_{\mathcal{M}}, \mathcal{D}_i), \tag{2}$$

where $L$ denotes the total number of layers. Throughout this work, we focus on this setting, where each model is compressed into low-rank form. The next section investigates how conventional merging strategies behave under low-rank setting.

## 4 MOTIVATING OBSERVATION

We now examine how conventional model merging strategies behave when applied to low-rank compressed models, as introduced in the previous section. For each layer $l$, two natural extensions of merging have been considered in prior work (Tang et al., 2025; Zhao et al., 2025; Zheng et al., 2025; Zhang et al., 2025; Stoica et al., 2025):

- **Combined merging:** $\hat{\theta}_{\mathcal{M}}^l = \theta_{\text{pre}}^l + \mathcal{M}(\{\boldsymbol{A}_i^l \boldsymbol{B}_i^l\}_{i=1}^n)$, where each low-rank update is first reconstructed into its full-rank form before merged.
- **Separate merging:** $\hat{\theta}_{\mathcal{M}}^l = \theta_{\text{pre}}^l + \mathcal{M}(\{\boldsymbol{A}_i^l\}_{i=1}^n) \mathcal{M}(\{\boldsymbol{B}_i^l\}_{i=1}^n)$, where the low-rank factors are merged independently before recomposition.

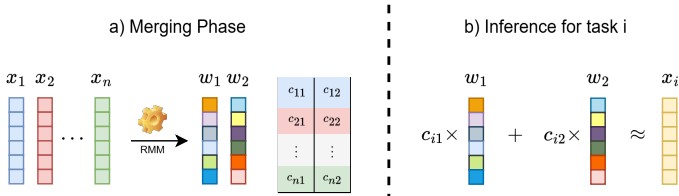

Figure 3: Overview of the proposed RMM framework. **Left:** During the merging phase, we compute a compact set of basis directions (e.g., two in this example) and their associated task-specific coefficients. **Right:** At inference, a target model $i$ is reconstructed by linearly combining the basis vectors using the corresponding coefficients.

While both approaches are theoretically valid, the combined strategy is highly inefficient in practice. Specifically, storing the merged update in combined form requires holding an $m \times d$ matrix. In contrast, the separate approach only stores an equivalent amount of the one of the low-rank factors $\boldsymbol{A}_i^l \in \mathbb{R}^{m \times r}$ and $\boldsymbol{B}_i^l \in \mathbb{R}^{r \times d}$. Since $r(m + d) \ll md$ for $r \ll \min(m, d)$, the storage and memory overhead of the combined approach defeats the purpose of compression. As an illustrative example, consider a layer with $m = d = 1024$ and a compression rank $r = 16$. The combined approach requires storing a full update of size $md = 1{,}048{,}576$ parameters while the separate approach requires storing $r(m + d) = 16 \times (1024 + 1024) = 32{,}768$ parameters.

This yields a **32× reduction in storage**, demonstrating that the combined approach entirely undermines the efficiency benefits of low-rank compression. Thus, we focus on the separate strategy as the only viable option for merging compressed models in practice.

We applied several existing model merging baselines to low-rank compressed models under varying compression ratios, using the separate approach described above. We selected RoBERTa-base (Zhuang et al., 2021) as the pre-trained model and conducted experiments on 5 GLUE tasks (Wang et al., 2018) (More details in Appendix A.4). As illustrated in Figures 1 and 2, merging compressed models leads to a *catastrophic drop* in average task performance compared to merging full-rank models. We attribute this degradation to two factors: (i) **Amplified task interference:** compression discards much of the representation capacity, leaving low-rank models more susceptible to destructive interference when merged; (ii) **Loss of coupling between $\boldsymbol{A}$ and $\boldsymbol{B}$:** conventional merging treats low-rank matrices $\boldsymbol{A}_i^l$ and $\boldsymbol{B}_i^l$ as independent factors, ignoring the coupled nature of their product, leading to structural misalignment in the merged updates. Together, these severely impair performance of merged low-rank models and suggest a deeper incompatibility between traditional merging methods and compressed representations.

This observation leads to a critical insight: under conventional strategies, no merged model can reliably recover the task-specific performance of the original low-rank adapters. This raises a natural question: *Can we instead reconstruct the original compressed models, or approximate them closely, by retaining a compact set of basis models, perhaps equivalent to two or more adapters depending on our storage budget?* Rather than forcing all updates into a single representation, we explore an alternative design space where merging constructs a *reconstruction-capable model space*. Next, we formalize this perspective and introduce our proposed method, *Reversible Model Merging* (RMM).

## 5 REVERSIBLE MODEL MERGING

The empirical failures of conventional merging in the compressed setting suggest a fundamental limitation: no single merged model can faithfully preserve the task-specific performance of all low-rank compressed models. Motivated by this, we propose a new approach, RMM, that departs from the goal of merging all models into a single set of weights. Instead, RMM constructs a small set of shared components (basis models) from which all original task models can be approximately reconstructed. Rather than forcing all $n$ task-specific models into one merged solution $\hat{\boldsymbol{\theta}}_{\mathcal{M}}$, RMM retains a compact set of $p$ merged components, where $2 \leq p \ll n$. These form a low-dimensional subspace that span the essential variations across tasks, enabling efficient recovery of each model on demand. The design balances performance and memory: by adjusting $p$, practitioners can trade off storage cost against reconstruction accuracy.

For each layer $l$, we focus on the $r$-dimensional vectors in low-rank delta matrices ($\boldsymbol{A}_i^l$ and $\boldsymbol{B}_i^l$), which are rows in $\boldsymbol{A}_i^l$ and columns in $\boldsymbol{B}_i^l$ ($r$ is the compression rank in compressed models). We treat each row of $\boldsymbol{A}_i^l$ and each column of $\boldsymbol{B}_i^l$ as an independent *task vector* and refer to a specific row or column index as the *task vector position*. For any fixed position (i.e., row or column index), we collect the corresponding task vectors across all $n$ models, and refer to the task vector from model $i$ at that position as $\boldsymbol{x}_i \in \mathbb{R}^r$. Our algorithm is then applied independently to each such position (i.e., a row index in $\boldsymbol{A}_i^l$ or column index in $\boldsymbol{B}_i^l$): we compress the set of task vectors $\{\boldsymbol{x}_i\}_{i=1}^n$ by projecting them into a shared basis, enabling later reconstruction as needed. We have a total of $m + d$ task vector positions per layer, $m$ rows from $\boldsymbol{A}_i^l$ and $d$ columns from $\boldsymbol{B}_i^l$, and the algorithm is applied separately to each of them.

For this purpose, we identify a set of basis vectors $\{\boldsymbol{w}_1, \ldots, \boldsymbol{w}_p\}$, where each $\boldsymbol{w}_j \in \mathbb{R}^r$ for $1 \leq j \leq p$. The span of these basis vectors defines a subspace $\boldsymbol{W} = [\boldsymbol{w}_1 \ \ldots \ \boldsymbol{w}_p] \in \mathbb{R}^{r \times p}$ into which each task vector $\boldsymbol{x}_i$ is projected, with the requirement that the projection reconstructs $\boldsymbol{x}_i$ as accurately as possible. We introduce $\boldsymbol{C} \in \mathbb{R}^{n \times p}$ as the coefficient matrix, where its $i$-th row $[c_{i1}, \ldots, c_{ip}]$ specifies the scalar weights used to linearly combine the basis vectors for reconstructing $\boldsymbol{x}_i$. The goal is then to determine $\boldsymbol{W}$ and $\boldsymbol{C}$ such that the sum of the reconstruction error across all the $n$ task vectors is minimized. This leads to the following optimization problem:

$$\min_{\boldsymbol{W}, \boldsymbol{C}} \ \sum_{i=1}^n \left\| \boldsymbol{x}_i - \sum_{j=1}^p \boldsymbol{c}_{ij} \boldsymbol{w}_j \right\|_2^2. \tag{3}$$

This formulation requires storing $rp$ parameters for the basis and $np$ for the coefficients, totaling $p(r + n)$, a significant reduction from the $rn$ parameters required to store all task vectors directly. Importantly, the value of $p$ is a tunable hyperparameter that controls the performance-storage trade-off. Figure 3 illustrates our approach for $p = 2$. The optimization in equation 3 can be rewritten in matrix form as follows:

$$\min_{\boldsymbol{W}, \boldsymbol{C}} \ \left\| \boldsymbol{X} - \boldsymbol{C} \boldsymbol{W}^\top \right\|_F^2, \tag{4}$$

where $\boldsymbol{X} = [\boldsymbol{x}_1 \ \ldots \boldsymbol{x}_n]^\top \in \mathbb{R}^{n \times r}$ is the matrix of all task vectors. We know that for any $\boldsymbol{W}$, the optimal $\boldsymbol{C}$ can be achieved by the least-squares solution, which is $\boldsymbol{C}^* = \boldsymbol{X} \boldsymbol{W} (\boldsymbol{W}^\top \boldsymbol{W})^{-1}$. Assuming that the basis vectors are orthonormal ($\boldsymbol{W}^\top \boldsymbol{W} = \boldsymbol{I}_p, p < r$), the optimization can be rewritten as:

$$\min_{\boldsymbol{W}} \ \left\| \boldsymbol{X} - \boldsymbol{X} \boldsymbol{W} \boldsymbol{W}^\top \right\|_F^2 \qquad \text{s.t.} \quad \boldsymbol{W}^\top \boldsymbol{W} = \boldsymbol{I}_p. \tag{5}$$

Note that in this formulation, we will store $\boldsymbol{W} \in \mathbb{R}^{r \times p}$ and $\boldsymbol{X} \boldsymbol{W} \in \mathbb{R}^{n \times p}$ ($\boldsymbol{X} \boldsymbol{W}$ is equivalent to $\boldsymbol{C}$ in equation 4).

**Theorem 1** (Optimal Solution to equation 5). *Let $\boldsymbol{X} \in \mathbb{R}^{n \times r}$ be zero-centered and $p < \min\{r, n\}$. The optimal basis $\boldsymbol{W}^*$ that minimizes equation 5 is given by the top-$p$ eigenvectors of the sample covariance matrix $\boldsymbol{X}^\top \boldsymbol{X}$. Equivalently, $\boldsymbol{W}^*$ consists of the top-$p$ right singular vectors of $\boldsymbol{X}$.*

The proof of Theorem 1 is provided in Appendix A.1. Next, we describe the two phases of RMM.

## 5.1 MERGING PHASE

We construct a compact representation of all task-specific models using a shared basis. For each layer and for each task vector position (a fixed row in $\boldsymbol{A}_i^l$ or column in $\boldsymbol{B}_i^l$), we gather the task vectors across all the $n$ models as $\boldsymbol{X} = [\boldsymbol{x}_1 \ \ldots \boldsymbol{x}_n]^\top \in \mathbb{R}^{n \times r}$. We first compute the mean vector $\boldsymbol{\mu} = \frac{1}{n} \sum_{i=1}^n \boldsymbol{x}_i$, and then zero-center $\boldsymbol{X}$ (i.e., $\boldsymbol{X} \leftarrow [\boldsymbol{x}_1 - \boldsymbol{\mu} \ \ldots \ \boldsymbol{x}_n - \boldsymbol{\mu}]^\top$). Next, we perform SVD on the centered matrix $\boldsymbol{X} = \boldsymbol{U} \boldsymbol{\Sigma} \boldsymbol{V}^\top$, and select the top-$p$ right singular vectors to form the optimal orthonormal basis $\boldsymbol{W}^*$ minimizing the reconstruction error in equation 5, i.e., $\boldsymbol{W}^* = \boldsymbol{V}_{[:,1:p]} \in \mathbb{R}^{r \times p}$. We then compute the corresponding coefficient matrix $\boldsymbol{C}^* = \boldsymbol{X} \boldsymbol{W}^* \in \mathbb{R}^{n \times p}$, storing the linear combination weights to reconstruct the original task vectors. The $i$-th row of $\boldsymbol{C}^*$, $[c_{i1}, \ldots, c_{ip}]$, contains the coefficients needed to reconstruct $\boldsymbol{x}_i$. At the end of this phase, we have discarded the original task vectors $\{\boldsymbol{x}_i\}_{i=1}^n$ and retained only the basis $\boldsymbol{W}^*$, the coefficients $\boldsymbol{C}^*$, and the mean vector $\boldsymbol{\mu}$ for each task vector position. In total, this representation achieves a memory-efficient compression from $rn$ parameters (to retain all task vectors) down to $p(r + n) + r$ (for each task vector position). We note that the merging phase is performed offline prior to inference.

## 5.2 INFERENCE PHASE

Following the common practice in model merging, we assume that the target task corresponding to each input is either explicitly specified by the user or automatically determined by an oracle router that assigns the input to the most suitable model index. Once the target task index $i$ has been identified, our objective is to reconstruct its task vector $x_i$, denoted by $\hat{x}_i$, using the compact representation $(W^*, C^*, \mu)$ obtained during the merging phase at each task vector position. We then recover $A_i^l$ and $B_i^l$ by concatenating all the reconstructed task vectors ($\hat{x}_i$ vectors) across all the positions (i.e., all the corresponding rows of $A_i^l$ or columns of $B_i^l$). Recall that for each task vector position, $W^* \in \mathbb{R}^{r \times p}$ encodes the orthonormal basis vectors, $C^* \in \mathbb{R}^{n \times p}$ stores the task-specific coefficients, and $\mu \in \mathbb{R}^r$ is the mean vector across all task vectors. The reconstruction is performed by linearly combining the basis directions using task-specific coefficients shifted with the mean vector $\mu$, i.e., $\hat{x}_i = \sum_{j=1}^{p} c_{ij}^* w_j^* + \mu = C_{[i,:]}^* W^{*\top} + \mu$. The resulting $\hat{x}_i \in \mathbb{R}^r$ thus serves as an approximation of the original task vector $x_i$ that was discarded during the merging phase. This reconstruction is performed independently for each task vector position (each row of $A_i^l$ and each column of $B_i^l$) and for all layers, thereby fully reconstructing the low-rank delta for task-specific model $i$, i.e., $\hat{\Delta}_i$. The complete procedure for the merging and inference phases is outlined in Algorithm 1 in Appendix A.2. Next, we present an empirical evaluation of our method and compare its performance against standard baselines.

# 6 EXPERIMENTS

To assess effectiveness of RMM, we conduct extensive experiments across a variety of compression settings, tasks and model architectures. Our goal is to evaluate whether RMM can preserve the performance of low-rank task-specific models while reducing the overall storage cost, and how it compares to conventional merging baselines.

## 6.1 EXPERIMENTAL SETUP

We first evaluate RMM on eight RoBERTa-base models fine-tuned for eight different tasks (Zhuang et al., 2021). Each model is individually compressed using either PT-SVD or LoRA with target ranks $r \in \{16, 32, 64, 128\}$. After compression, we apply RMM to the resulting low-rank models and compare its performance against standard merging baselines: Task Arithmetic (TA) (Ilharco et al., 2023), TIES-merging (Yadav et al., 2023), and DARE (Yu et al., 2024). As an upper bound, we also report the performance of individual compressed models before merging. Additional results are provided in Appendix A.3, including experiments on OPT-1.3b (Zhang et al., 2022) and on ViT-B/32 (Radford et al., 2021) for vision tasks, with different numbers of tasks. Following the GLUE benchmark protocol (Wang et al., 2018), we evaluate each task using its standard metric. QNLI (question–answering inference), MRPC (paraphrase detection), SST-2 (sentiment classification), MNLI (multi-genre natural language inference), QQP (duplicate question detection), and RTE (textual entailment) are measured by accuracy. CoLA (linguistic acceptability) is evaluated using Matthews correlation, while STS-B (semantic textual similarity) is assessed with Pearson correlation. We report the average score across all tasks as an overall performance measure. We also compute the relative storage cost for each method measured as the fraction of parameters retained relative to storing all individual task-specific adapters. Baseline methods that store a single merged model, have a fixed cost of $\frac{r}{rn} = \frac{1}{n}$. For RMM with $p$ basis components, the relative storage cost is given by $\frac{p(r+n)+r}{rn}$, as derived in Section 5.1.

## 6.2 RESULTS AND DISCUSSION

Tables 1 and 2 summarize the performance of RMM compared to baselines under PT-SVD and LoRA compression, respectively. Across both settings and for all target ranks $r \in \{16, 32, 64, 128\}$, our method consistently outperforms TA, TIES, and DARE by a large margin. At lower ranks ($r = 16, 32$), baseline methods fail catastrophically, achieving only 31–42% average score and performing particularly poorly on structurally sensitive tasks such as CoLA and STS-B. In contrast, RMM with $p = 2$ achieves substantial gains (e.g., 57.24% under PT-SVD and 51.41% under LoRA at $r = 16$), while RMM with $p = 3$ delivers even stronger results (up to 72.22% and 58.64% at

Table 1: Performance on GLUE benchmark for merging eight RoBERTa-base models compressed with PT-SVD at various ranks ($r = 16, 32, 64, 128$).

| $r$ | Method | Storage | QNLI | MRPC | SST-2 | MNLI | QQP | RTE | COLA | STSB | Average |
|---|---|---|---|---|---|---|---|---|---|---|---|
| 16 | No merging | 100% | 92.02 | 85.68 | 93.69 | 85.21 | 87.19 | 77.98 | 58.05 | 90.59 | 83.80 |
| | TA | 13% | 50.59 | 33.51 | 49.08 | 31.82 | 36.82 | 47.29 | 0.00 | 4.19 | 31.66 |
| | TIES | 13% | 50.56 | 34.78 | 49.08 | 31.82 | 36.82 | 47.29 | 0.00 | 3.75 | 31.76 |
| | DARE | 13% | 50.80 | 33.51 | 49.08 | 31.82 | 36.82 | 47.29 | 0.00 | 5.07 | 31.80 |
| | **RMM** ($p = 2$) | 50% | 51.20 | 78.20 | 88.30 | 66.74 | 86.02 | 47.29 | 6.56 | 33.59 | 57.24 |
| | **RMM** ($p = 3$) | 69% | 88.83 | 86.26 | 92.89 | 79.58 | 87.18 | 48.74 | 33.85 | 60.46 | 72.22 |
| 32 | No merging | 100% | 92.31 | 85.86 | 94.15 | 86.38 | 88.62 | 78.34 | 59.33 | 90.68 | 84.46 |
| | TA | 13% | 50.59 | 33.51 | 49.08 | 31.82 | 36.82 | 47.29 | 0.00 | 4.21 | 31.66 |
| | TIES | 13% | 50.56 | 35.19 | 49.08 | 31.82 | 36.82 | 47.29 | 0.00 | 3.87 | 31.83 |
| | DARE | 13% | 50.72 | 33.51 | 49.08 | 31.82 | 36.82 | 47.29 | 0.00 | 3.48 | 31.59 |
| | **RMM** ($p = 2$) | 44% | 50.54 | 74.32 | 55.05 | 62.68 | 87.07 | 47.29 | 0.00 | 23.18 | 50.02 |
| | **RMM** ($p = 3$) | 59% | 84.66 | 86.32 | 92.89 | 79.70 | 88.05 | 47.29 | 23.57 | 44.54 | 68.38 |
| 64 | No merging | 100% | 92.51 | 86.03 | 94.38 | 87.18 | 90.15 | 78.70 | 61.11 | 90.72 | 85.10 |
| | TA | 13% | 50.61 | 33.51 | 49.08 | 31.82 | 36.82 | 47.29 | 0.00 | 4.23 | 31.67 |
| | TIES | 13% | 50.58 | 35.13 | 49.08 | 31.82 | 36.82 | 47.29 | 0.00 | 3.80 | 31.82 |
| | DARE | 13% | 50.58 | 33.51 | 49.08 | 31.82 | 36.82 | 47.29 | 0.00 | 5.22 | 31.79 |
| | **RMM** ($p = 2$) | 41% | 50.53 | 73.10 | 49.08 | 58.48 | 86.98 | 47.29 | 0.00 | 17.50 | 47.87 |
| | **RMM** ($p = 3$) | 55% | 74.14 | 86.20 | 92.43 | 78.83 | 89.00 | 47.29 | 13.16 | 33.70 | 64.34 |
| 128 | No merging | 100% | 92.68 | 86.14 | 94.50 | 87.33 | 91.05 | 79.78 | 60.16 | 90.74 | 85.30 |
| | TA | 13% | 50.59 | 33.51 | 49.08 | 31.82 | 36.82 | 47.29 | 0.00 | 4.23 | 31.67 |
| | TIES | 13% | 50.58 | 35.54 | 49.08 | 31.82 | 36.82 | 47.29 | 0.00 | 3.93 | 31.88 |
| | DARE | 13% | 50.61 | 34.78 | 49.08 | 31.82 | 36.82 | 47.29 | 0.00 | 2.41 | 31.60 |
| | **RMM** ($p = 2$) | 39% | 50.54 | 70.78 | 49.08 | 54.69 | 84.68 | 47.29 | 0.00 | 14.53 | 46.45 |
| | **RMM** ($p = 3$) | 52% | 59.20 | 85.80 | 92.32 | 77.50 | 89.02 | 47.29 | 6.56 | 26.94 | 60.58 |

Table 2: Performance on GLUE benchmark for merging eight RoBERTa-base models compressed with LoRA at various ranks ($r = 16, 32, 64, 128$).

| $r$ | Method | Storage | QNLI | MRPC | SST-2 | MNLI | QQP | RTE | COLA | STSB | Average |
|---|---|---|---|---|---|---|---|---|---|---|---|
| 16 | No merging | 100% | 61.38 | 83.48 | 94.50 | 83.90 | 87.56 | 68.23 | 53.92 | 88.71 | 77.71 |
| | TA | 13% | 50.59 | 33.51 | 72.13 | 34.79 | 64.46 | 47.29 | 0.00 | 21.81 | 40.57 |
| | TIES | 13% | 50.63 | 33.68 | 74.31 | 35.64 | 67.45 | 47.29 | 0.00 | 26.38 | 41.92 |
| | DARE | 13% | 50.63 | 33.51 | 73.74 | 35.97 | 65.35 | 47.29 | 0.00 | 21.62 | 41.01 |
| | KnOTS-TIES | 13% | 50.63 | 33.51 | 69.72 | 35.42 | 65.99 | 47.29 | 0.00 | 27.62 | 41.27 |
| | **RMM** ($p = 2$) | 50% | 51.13 | 34.14 | 90.71 | 60.48 | 84.41 | 47.29 | 0.00 | 43.14 | 51.41 |
| | **RMM** ($p = 3$) | 69% | 55.92 | 38.14 | 93.12 | 82.95 | 87.30 | 47.29 | 6.56 | 57.81 | 58.64 |
| 32 | No merging | 100% | 60.13 | 81.28 | 93.81 | 84.46 | 88.18 | 62.82 | 53.12 | 88.53 | 76.54 |
| | TA | 13% | 50.91 | 68.06 | 54.70 | 33.77 | 65.50 | 55.96 | 0.00 | -0.48 | 41.05 |
| | TIES | 13% | 51.03 | 68.06 | 56.77 | 34.24 | 66.27 | 58.12 | 0.00 | 1.02 | 41.94 |
| | DARE | 13% | 50.96 | 64.64 | 63.07 | 35.12 | 63.66 | 55.60 | 0.00 | 1.01 | 41.76 |
| | KnOTS-TIES | 13% | 51.16 | 68.29 | 54.47 | 34.77 | 65.25 | 58.48 | 0.00 | 1.59 | 41.75 |
| | **RMM** ($p = 2$) | 44% | 52.08 | 65.39 | 81.54 | 62.46 | 77.06 | 57.40 | 0.00 | 9.95 | 50.73 |
| | **RMM** ($p = 3$) | 59% | 54.62 | 69.97 | 92.20 | 81.29 | 85.97 | 57.40 | 4.64 | 24.08 | 58.77 |
| 64 | No merging | 100% | 59.62 | 84.12 | 94.15 | 85.20 | 88.75 | 67.15 | 58.31 | 88.70 | 78.25 |
| | TA | 13% | 54.75 | 68.52 | 77.29 | 37.24 | 36.82 | 47.29 | 0.00 | -3.53 | 39.80 |
| | TIES | 13% | 54.82 | 68.75 | 80.73 | 37.67 | 36.82 | 47.29 | 0.00 | -2.48 | 40.45 |
| | DARE | 13% | 53.71 | 68.58 | 78.56 | 38.33 | 36.82 | 47.29 | 0.00 | -4.28 | 39.88 |
| | KnOTS-TIES | 13% | 54.48 | 68.93 | 80.96 | 37.62 | 36.82 | 47.29 | 0.00 | -2.6 | 40.44 |
| | **RMM** ($p = 2$) | 41% | 55.74 | 69.28 | 87.27 | 42.81 | 77.72 | 47.29 | 0.00 | 4.91 | 48.13 |
| | **RMM** ($p = 3$) | 55% | 59.07 | 70.14 | 90.83 | 79.11 | 82.06 | 47.29 | 0.00 | 14.70 | 55.40 |
| 128 | No merging | 100% | 61.58 | 85.04 | 93.58 | 85.71 | 89.68 | 57.04 | 59.35 | 89.68 | 77.71 |
| | TA | 13% | 51.49 | 69.22 | 54.47 | 37.16 | 63.27 | 47.29 | 0.00 | -1.74 | 40.15 |
| | TIES | 13% | 51.91 | 68.93 | 57.34 | 37.81 | 63.36 | 47.29 | 0.00 | -0.25 | 40.80 |
| | DARE | 13% | 51.97 | 68.99 | 58.60 | 35.64 | 63.33 | 47.29 | 0.00 | -0.44 | 40.67 |
| | KnOTS-TIES | 13% | 51.99 | 69.1 | 57.57 | 38.3 | 63.49 | 47.29 | 0.00 | 1.98 | 41.22 |
| | **RMM** ($p = 2$) | 39% | 57.18 | 67.83 | 88.53 | 49.54 | 84.64 | 47.29 | 0.00 | 28.00 | 52.88 |
| | **RMM** ($p = 3$) | 52% | 59.13 | 39.19 | 92.09 | 80.08 | 86.39 | 47.29 | 15.45 | 71.18 | 61.35 |

$r = 16$), significantly reducing the gap to individual compressed models. At higher ranks ($r = 64, 128$), where compression error is reduced, RMM continues to dominate the baselines. With $p = 3$, RMM achieves $64.34\%$ and $60.58\%$ with PT-SVD, and $55.40\%$ and $61.35\%$ with LoRA, consistently surpassing single-model baselines, which remain stuck near $40\%$. Even RMM with $p = 2$, maintains clear advantages, confirming that reversible approach prevents the destructive interference that degrades performance in prior methods. We also observe that as the rank increases,

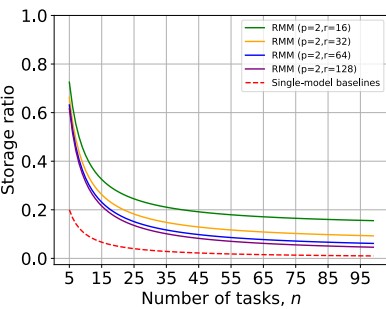

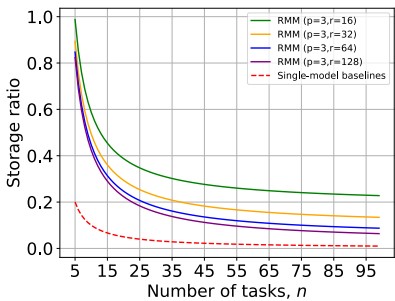

Figure 4: Scalability of RMM ($p = 2$). The storage ratio indicates the percentage of memory required relative to storing all models.

Figure 5: Scalability of RMM ($p = 3$). The storage ratio indicates the percentage of memory required relative to storing all models.

the relative gain of RMM over baselines becomes smaller. One plausible explanation is that higher-rank task representations lie in higher-dimensional spaces and become more expressive and less aligned, making it harder to compress them into a shared low-dimensional basis. Overall, the results demonstrate that RMM provides a consistent advantage across both compression regimes and all ranks: it preserves high task performance while incurring some extra storage which offers a tunable trade-off between accuracy and storage efficiency by adjusting the number of reversible components. Importantly, this storage cost pertains to **offline memory**, typically CPU-based and inexpensive, whereas task performance directly impacts runtime quality. In such cases, the slight increase in CPU-side storage is often a worthwhile trade-off for significantly better accuracy.

### 6.3 SCALABILITY OF RMM

We further examine the scalability of RMM as the number of models, $n$, increases. Figure 4 and 5 illustrate the storage ratio (relative to storing all individual task-specific models) as a function of the number of tasks for RMM with $p = 2$ and $p = 3$, respectively. The results confirm that RMM scales favorably: as the number of tasks grows, the relative storage cost decreases significantly. This trend highlights the key benefit of reversible merging: storage grows sublinearly with the number of task-specific models. This sublinear growth is a direct consequence of RMM design: once the basis is fixed, adding new tasks only requires storing an additional row in the coefficient matrix. In contrast, traditional approaches require either storing one merged model (which sacrifices performance) or storing each task-specific model independently (which scales linearly). These findings highlight the practical utility of RMM in large-scale multi-task, continual or federated settings, where it may be necessary to manage hundreds of models efficiently without compromising task performance.

## 7 CONCLUSION

In this work, we introduced a principled framework for reversible model merging that redefines how multiple fine-tuned models can be stored, merged, and reconstructed. Instead of forcing all task-specific knowledge into a single merged model, we relax this constraint by storing a compact set of intermediate representations—substantially fewer than the total number of original models, yet more flexible than a single aggregate. This design enables accurate reconstruction of task-specific models while significantly reducing storage overhead. Our method offers a controllable trade-off between performance and storage efficiency, especially in the context of low-rank compressed models. By representing fine-tuned models through compact bases and reversible mappings, we show that it is possible to merge, reconstruct, and deploy models with high accuracy while adhering to strict memory budgets. This flexibility enables practitioners to adapt the degree of compression and reconstruction fidelity to the demands of their application. Overall, our framework moves beyond heuristic or one-shot merging techniques by formalizing model merging as a structured, reversible process. This new perspective not only broadens the design space of efficient multi-task deployment but also opens promising directions for future research in performance-aware compression and reconstruction strategies.

## Reproducibility Statement

For models compressed with PT-SVD, we performed truncated SVD on fully fine-tuned models. For LoRA-compressed models, we used the peft library to perform LoRA fine-tuning (Mangrulkar et al., 2022). To report results for baseline methods, we applied their code to low-rank weights. Our implementation of RMM, along with all required checkpoints, will be publicly available to facilitate the reproducibility of our results.

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

# A APPENDIX

## A.1 PROOF OF THEOREM 1

Let $\boldsymbol{A} = \boldsymbol{X} - \boldsymbol{X}\boldsymbol{W}\boldsymbol{W}^\top$. By definition of the Frobenius norm, $\|\boldsymbol{A}\|_F^2 = \mathrm{Tr}(\boldsymbol{A}^\top \boldsymbol{A})$, where $\mathrm{Tr}(\cdot)$ denotes trace of a square matrix. Expanding gives:

$$\|\boldsymbol{X} - \boldsymbol{X}\boldsymbol{W}\boldsymbol{W}^\top\|_F^2 = \mathrm{Tr}\big((\boldsymbol{X} - \boldsymbol{X}\boldsymbol{W}\boldsymbol{W}^\top)^\top(\boldsymbol{X} - \boldsymbol{X}\boldsymbol{W}\boldsymbol{W}^\top)\big).$$

We expand the term in $\mathrm{Tr}()$:

$$\begin{aligned}
(\boldsymbol{X} - \boldsymbol{X}\boldsymbol{W}\boldsymbol{W}^\top)^\top(\boldsymbol{X} - \boldsymbol{X}\boldsymbol{W}\boldsymbol{W}^\top) &= (\boldsymbol{X}^\top - \boldsymbol{W}\boldsymbol{W}^\top\boldsymbol{X}^\top)(\boldsymbol{X} - \boldsymbol{X}\boldsymbol{W}\boldsymbol{W}^\top) \\
&= \boldsymbol{X}^\top\boldsymbol{X} - \boldsymbol{X}^\top\boldsymbol{X}\boldsymbol{W}\boldsymbol{W}^\top - \boldsymbol{W}\boldsymbol{W}^\top\boldsymbol{X}^\top\boldsymbol{X} \\
&\quad + \boldsymbol{W}\boldsymbol{W}^\top\boldsymbol{X}^\top\boldsymbol{X}\boldsymbol{W}\boldsymbol{W}^\top.
\end{aligned}$$

Next, we take the trace of the expanded form. Using cyclicity of trace and $\boldsymbol{W}^\top\boldsymbol{W} = \boldsymbol{I}_p$:

$$\begin{aligned}
\|\boldsymbol{X} - \boldsymbol{X}\boldsymbol{W}\boldsymbol{W}^\top\|_F^2 &= \mathrm{Tr}(\boldsymbol{X}^\top\boldsymbol{X}) - 2\mathrm{Tr}(\boldsymbol{W}^\top\boldsymbol{X}^\top\boldsymbol{X}\boldsymbol{W}) + \mathrm{Tr}(\boldsymbol{W}^\top\boldsymbol{W}\boldsymbol{W}^\top\boldsymbol{X}^\top\boldsymbol{X}\boldsymbol{W}) \\
&= \mathrm{Tr}(\boldsymbol{X}^\top\boldsymbol{X}) - 2\mathrm{Tr}(\boldsymbol{W}^\top\boldsymbol{X}^\top\boldsymbol{X}\boldsymbol{W}) + \mathrm{Tr}(\boldsymbol{W}^\top\boldsymbol{X}^\top\boldsymbol{X}\boldsymbol{W}) \\
&= \mathrm{Tr}(\boldsymbol{X}^\top\boldsymbol{X}) - \mathrm{Tr}(\boldsymbol{W}^\top\boldsymbol{X}^\top\boldsymbol{X}\boldsymbol{W}).
\end{aligned}$$

Since $\mathrm{Tr}(\boldsymbol{X}^\top\boldsymbol{X})$ is a constant, minimizing $\|\boldsymbol{X} - \boldsymbol{X}\boldsymbol{W}\boldsymbol{W}^\top\|_F^2$ is equivalent to maximizing $\mathrm{Tr}(\boldsymbol{W}^\top\boldsymbol{X}^\top\boldsymbol{X}\boldsymbol{W})$. Hence,

$$\min_{\boldsymbol{W}^\top\boldsymbol{W} = \boldsymbol{I}_p} \|\boldsymbol{X} - \boldsymbol{X}\boldsymbol{W}\boldsymbol{W}^\top\|_F^2 \iff \max_{\boldsymbol{W}^\top\boldsymbol{W} = \boldsymbol{I}_p} \mathrm{Tr}(\boldsymbol{W}^\top\boldsymbol{X}^\top\boldsymbol{X}\boldsymbol{W}).$$

By the Rayleigh quotient theorem (Horn & Johnson, 2012), the maximizer is obtained by taking the columns of $\boldsymbol{W}$ to be the top-$p$ eigenvectors of $\boldsymbol{X}^\top\boldsymbol{X}$. This is precisely the principal component analysis (PCA) solution: it finds the $p$-dimensional subspace that maximizes the projected variance and, equivalently, minimizes the squared reconstruction error. Moreover, if $\boldsymbol{X} = \boldsymbol{U}\boldsymbol{\Sigma}\boldsymbol{V}^\top$ is the SVD of $\boldsymbol{X}$, then $\boldsymbol{X}^\top\boldsymbol{X} = \boldsymbol{V}\boldsymbol{\Sigma}^2\boldsymbol{V}^\top$, so the eigenvectors of $\boldsymbol{X}^\top\boldsymbol{X}$ are precisely the right singular vectors (columns of $\boldsymbol{V}$). Hence, the PCA solution can also be obtained by taking the top-$p$ right singular vectors of $\boldsymbol{X}$. In practice, this SVD-based approach is numerically more stable and often computationally preferable to directly forming $\boldsymbol{X}^\top\boldsymbol{X}$, especially when $r$ is large.

## A.2 RMM ALGORITHM

---

**Algorithm 1** Reversible Model Merging (RMM)

---

1: **Merging Phase:**
2: Require: Low-rank deltas $A_i^l$ and $B_i^l$, compression rank $r$, number of basis vectors $p$
3: **for** $l = 1$ to $L$ **do**
4:     **for** each task vector position (each row in $A_i^l$ or each column in $B_i^l$) **do**
5:         Gather task vectors $x_i$ of each model $i$ ($1 \leq i \leq n$)
6:         $X \leftarrow [x_1 \ \ldots \ x_n]^\top \in \mathbb{R}^{n \times r}$
7:         $\mu \leftarrow \sum_{i=1}^n x_i \in \mathbb{R}^r$
8:         $X \leftarrow [x_1 - \mu \ \ldots \ x_n - \mu]^\top$
9:         $U, \Sigma, V^\top \leftarrow \text{SVD}(X)$
10:       $W^* \leftarrow V_{[:,1:p]} \in \mathbb{R}^{r \times p}$
11:       $C^* \leftarrow X W^* \in \mathbb{R}^{n \times p}$
12:       Store $(W^*, C^*, \mu)$ and discard task vectors $\{x_1 \ \ldots \ x_n\}$
13:     **end for**
14: **end for**
15: **Inference Phase:**
16: Require: target task $i$, stored representation $(W^*, C^*, \mu)$ for each task vector position
17: **for** $l = 1$ to $L$ **do**
18:     **for** each task vector position **do**
19:         $\hat{x}_i = C^*_{[i,:]} W^{*\top} + \mu$
20:     **end for**
21:     $\hat{A}_i^l = \text{Concatenate}(\hat{x}_i \text{ vectors across all row positions})$
22:     $\hat{B}_i^l = \text{Concatenate}(\hat{x}_i \text{ vectors across all column positions})$
23: **end for**
24: $\hat{\Delta}_i = \{\hat{A}_i^l \hat{B}_i^l\}_{l=1}^L$
25: Inference with the reconstructed model $\theta_{pre} + \hat{\Delta}_i$

---

| $r$ | Method | Storage | QNLI | MRPC | SST-2 | MNLI | QQP | BOOLQ | Average |
|---|---|---|---|---|---|---|---|---|---|
| | No merging | 100% | 91.74 | 71.54 | 95.30 | 84.19 | 88.14 | 71.47 | 83.73 |
| | Task-Arithmetic | 17% | 50.34 | 37.16 | 58.49 | 34.43 | 63.30 | 41.87 | 47.60 |
| 16 | TIES | 17% | 49.79 | 36.75 | 61.24 | 35.57 | 63.55 | 40.67 | 47.93 |
| | DARE | 17% | 49.95 | 37.68 | 56.08 | 34.51 | 63.39 | 42.78 | 47.40 |
| | **RMM** ($p = 2$) | 63% | 88.12 | 44.00 | 93.58 | 73.06 | 85.63 | 59.69 | 74.01 |
| | **RMM** ($p = 3$) | 85% | 91.58 | 49.04 | 95.64 | 80.31 | 87.52 | 65.50 | 78.27 |
| | No merging | 100% | 92.06 | 72.17 | 95.41 | 84.60 | 89.17 | 72.45 | 84.31 |
| | Task-Arithmetic | 17% | 50.28 | 37.22 | 58.94 | 34.57 | 63.33 | 41.93 | 47.71 |
| 32 | TIES | 17% | 49.77 | 36.93 | 61.93 | 35.69 | 63.65 | 40.73 | 48.12 |
| | DARE | 17% | 50.39 | 40.87 | 56.19 | 34.96 | 63.59 | 43.33 | 48.22 |
| | **RMM** ($p = 2$) | 56% | 86.66 | 41.91 | 91.74 | 71.75 | 85.68 | 56.39 | 72.36 |
| | **RMM** ($p = 3$) | 76% | 91.85 | 44.81 | 94.84 | 80.21 | 88.43 | 63.52 | 77.28 |
| | No merging | 100% | 92.26 | 72.12 | 95.41 | 84.64 | 90.07 | 72.84 | 84.56 |
| | Task-Arithmetic | 17% | 50.28 | 37.28 | 59.06 | 34.56 | 63.34 | 42.08 | 47.77 |
| 64 | TIES | 17% | 49.75 | 37.04 | 61.81 | 35.84 | 63.74 | 40.76 | 48.16 |
| | DARE | 17% | 50.16 | 40.52 | 58.83 | 35.59 | 63.35 | 42.54 | 48.50 |
| | **RMM** ($p = 2$) | 53% | 83.87 | 42.72 | 89.45 | 69.76 | 85.70 | 53.43 | 70.82 |
| | **RMM** ($p = 3$) | 71% | 92.07 | 41.97 | 94.50 | 79.70 | 88.73 | 61.96 | 76.49 |
| | No merging | 100% | 92.35 | 72.41 | 95.41 | 84.67 | 90.93 | 73.09 | 84.81 |
| | Task-Arithmetic | 17% | 50.32 | 37.39 | 58.94 | 34.63 | 63.37 | 42.08 | 47.79 |
| 128 | TIES | 17% | 49.79 | 37.04 | 62.04 | 35.99 | 63.74 | 40.55 | 48.19 |
| | DARE | 17% | 50.03 | 38.14 | 57.91 | 34.72 | 63.31 | 42.11 | 47.70 |
| | **RMM** ($p = 2$) | 52% | 81.38 | 42.78 | 84.75 | 67.66 | 83.22 | 51.99 | 68.63 |
| | **RMM** ($p = 3$) | 69% | 92.28 | 41.80 | 93.69 | 79.42 | 88.76 | 59.69 | 75.94 |

Table 3: Performance on GLUE benchmark for merging six OPT-1.3b models compressed with PT-SVD at various ranks ($r = 16, 32, 64, 128$).

## A.3 ADDITIONAL RESULTS

In this section, we present extended results for two additional pre-trained models: OPT-1.3b and ViT-B/32. For **OPT-1.3b**, we evaluate on six natural language understanding tasks drawn from GLUE and related benchmarks: QNLI, MRPC, SST-2, MNLI, QQP, and BoolQ (Clark et al., 2019). For

| r | Method | Storage | MNIST | GTSRB | SVHN | Cars | EuroSAT | DTD | SUN397 | Average |
|---|--------|---------|-------|-------|------|------|---------|-----|--------|---------|
| 16 | No merging | 100% | 99.61 | 97.22 | 97.06 | 69.20 | 99.67 | 75.21 | 72.72 | 87.24 |
| | Task-Arithmetic | 14% | 47.19 | 28.29 | 38.64 | 56.27 | 57.52 | 41.38 | 62.92 | 47.46 |
| | TIES | 14% | 60.14 | 34.52 | 49.03 | 58.55 | 61.70 | 43.67 | 62.87 | 52.93 |
| | DARE | 14% | 53.80 | 31.00 | 44.96 | 57.60 | 56.93 | 43.99 | 62.42 | 50.10 |
| | **RMM** ($p = 2$) | 55% | 99.16 | 88.52 | 94.89 | 66.61 | 97.44 | 60.64 | 67.86 | 82.16 |
| | **RMM** ($p = 3$) | 76% | 99.58 | 95.65 | 96.54 | 68.30 | 98.93 | 67.18 | 69.50 | 85.10 |
| 32 | No merging | 100% | 99.65 | 98.45 | 97.35 | 73.04 | 99.78 | 78.72 | 75.52 | 88.93 |
| | Task-Arithmetic | 14% | 47.90 | 28.31 | 38.68 | 56.24 | 57.52 | 41.38 | 63.00 | 47.58 |
| | TIES | 14% | 61.25 | 35.28 | 49.95 | 58.80 | 62.41 | 44.04 | 63.05 | 53.54 |
| | DARE | 14% | 53.05 | 31.04 | 46.34 | 58.54 | 57.74 | 42.87 | 62.87 | 50.35 |
| | **RMM** ($p = 2$) | 49% | 98.98 | 88.23 | 94.29 | 68.32 | 96.81 | 61.91 | 68.36 | 82.41 |
| | **RMM** ($p = 3$) | 67% | 99.50 | 96.53 | 96.68 | 70.90 | 98.93 | 70.43 | 71.23 | 86.31 |
| 64 | No merging | 100% | 99.67 | 98.68 | 97.41 | 75.76 | 99.85 | 79.15 | 77.73 | 89.75 |
| | Task-Arithmetic | 14% | 47.99 | 28.30 | 38.71 | 56.21 | 57.59 | 41.38 | 63.22 | 47.63 |
| | TIES | 14% | 61.83 | 35.93 | 50.28 | 59.08 | 63.07 | 44.15 | 63.17 | 53.93 |
| | DARE | 14% | 54.78 | 32.79 | 45.05 | 58.23 | 61.30 | 42.87 | 62.77 | 51.11 |
| | **RMM** ($p = 2$) | 46% | 98.46 | 84.43 | 92.83 | 69.36 | 95.59 | 62.39 | 69.77 | 81.83 |
| | **RMM** ($p = 3$) | 62% | 99.45 | 96.14 | 96.33 | 72.84 | 98.56 | 72.07 | 73.85 | 87.03 |
| 128 | No merging | 100% | 99.68 | 98.75 | 97.46 | 76.99 | 99.85 | 79.04 | 78.59 | 90.05 |
| | Task-Arithmetic | 14% | 48.00 | 28.31 | 38.73 | 56.21 | 57.63 | 41.44 | 63.35 | 47.67 |
| | TIES | 14% | 62.08 | 36.36 | 50.48 | 59.23 | 63.41 | 44.68 | 63.25 | 54.21 |
| | DARE | 14% | 55.16 | 32.07 | 45.47 | 58.55 | 61.85 | 43.09 | 62.85 | 51.29 |
| | **RMM** ($p = 2$) | 44% | 97.22 | 78.56 | 90.63 | 70.22 | 94.00 | 61.49 | 71.03 | 80.45 |
| | **RMM** ($p = 3$) | 59% | 99.28 | 94.59 | 95.95 | 74.24 | 98.19 | 71.86 | 76.02 | 87.16 |

Table 4: Performance on vision tasks for merging seven ViT-B/32 models compressed with PT-SVD at various ranks ($r = 16, 32, 64, 128$).

**ViT-B/32**, we focus on vision classification tasks, covering seven diverse datasets: MNIST (LeCun et al., 1998), GTSRB (Stallkamp et al., 2011), SVHN (Netzer et al., 2011), Cars (Krause et al., 2013), EuroSAT (Helber et al., 2019), DTD (Cimpoi et al., 2014), and SUN397 (Xiao et al., 2010).

Table 3 reports the results for six OPT-1.3b models compressed with PT-SVD. Both variants of RMM ($p = 2$ and $p = 3$) consistently outperform the baselines, with RMM at $p = 3$ narrowing the gap to the original individual models while requiring substantially fewer parameters. Table 4 presents the results for seven ViT-B/32 models. In vision tasks, the benefit of RMM is even more pronounced: with $p = 3$, our method achieves an average accuracy less than $3\%$ below that of the original models, while reducing storage to only $59\%$ of that required to retain all models at $r = 128$. These results demonstrate that RMM can faithfully reconstruct the original compressed models while offering significant storage efficiency.

## A.4 Results for 5 tasks

For the experiments in Section 4, we used the RoBERTa-base model as the pretrained model and evaluated model merging baselines on five GLUE tasks (QNLI, MRPC, SST-2, MNLI, QQP). Figures 1 and 2 in Section 4 summarize these results. For completeness, Table 5 corresponds to Figure 1 and reports results for LoRA fine-tuned models, along with improvements from our method, and Table 6 corresponds to Figure 2 and reports results for PT-SVD compressed models.

## A.5 Inference latency

RMM introduces no additional overhead during inference beyond a one-time reconstruction step. Because the stored bases and per-task coefficients are independent across all task-vector positions, each reconstruction reduces to a set of independent low-rank matrix multiplications. These operations are highly amenable to GPU parallelism, allowing modern hardware to compute all required products concurrently. As a result, the reconstruction step is fast in practice, and its latency is negligible relative to the overall inference cost. Moreover, when sufficient GPU memory is available, multiple task-specific models can be reconstructed in parallel. This enables a distributed deployment regime where task-specialized weights are materialized once and reused across many queries, eliminating the need to reconfigure or recompose the model for each new task input.

| r | Method | Storage | QNLI | MRPC | SST-2 | MNLI | QQP | Average |
|---|---|---|---|---|---|---|---|---|
| | No merging | 100% | 61.38 | 83.48 | 94.50 | 83.90 | 87.56 | 82.16 |
| | Task-Arithmetic | 20% | 50.61 | 33.51 | 73.85 | 34.23 | 64.58 | 51.36 |
| 16 | TIES | 20% | 50.65 | 33.51 | 75.57 | 35.83 | 66.97 | 52.51 |
| | DARE | 20% | 50.61 | 33.51 | 79.36 | 34.70 | 63.40 | 52.32 |
| | **RMM** ($p = 2$) | 73% | 59.64 | 68.64 | 93.00 | 82.68 | 86.89 | 78.17 |
| | No merging | 100% | 60.13 | 81.28 | 93.81 | 84.46 | 88.18 | 81.57 |
| | Task-Arithmetic | 20% | 51.18 | 67.30 | 58.83 | 34.33 | 68.56 | 56.04 |
| 32 | TIES | 20% | 51.36 | 68.17 | 59.52 | 34.76 | 71.46 | 57.05 |
| | DARE | 20% | 50.83 | 64.87 | 68.12 | 35.54 | 68.93 | 57.66 |
| | **RMM** ($p = 2$) | 66% | 57.72 | 67.54 | 92.55 | 81.80 | 86.13 | 77.15 |
| | No merging | 100% | 59.62 | 84.12 | 94.15 | 85.20 | 88.75 | 82.37 |
| | Task-Arithmetic | 20% | 54.74 | 68.93 | 80.28 | 35.70 | 36.82 | 55.29 |
| 64 | TIES | 20% | 55.26 | 68.93 | 78.78 | 36.76 | 36.82 | 55.31 |
| | DARE | 20% | 54.71 | 68.81 | 74.54 | 38.72 | 36.82 | 54.72 |
| | **RMM** ($p = 2$) | 63% | 59.82 | 67.48 | 91.51 | 83.21 | 84.41 | 77.29 |
| | No merging | 100% | 61.58 | 85.04 | 93.58 | 85.71 | 89.68 | 83.12 |
| | Task-Arithmetic | 20% | 52.52 | 68.70 | 54.82 | 37.83 | 63.46 | 55.47 |
| 128 | TIES | 20% | 52.55 | 69.04 | 57.80 | 38.91 | 63.72 | 56.40 |
| | DARE | 20% | 55.04 | 69.62 | 54.47 | 38.92 | 64.12 | 56.43 |
| | **RMM** ($p = 2$) | 62% | 60.21 | 34.96 | 91.86 | 78.59 | 86.65 | 70.45 |

Table 5: Performance on GLUE benchmark for merging five RoBERTa-base models compressed with LoRA at various ranks ($r = 16, 32, 64, 128$).

| r | Method | Storage | QNLI | MRPC | SST-2 | MNLI | QQP | Average |
|---|---|---|---|---|---|---|---|---|
| | No merging | 100% | 92.02 | 85.68 | 93.69 | 85.21 | 87.19 | 88.76 |
| | Task-Arithmetic | 20% | 53.03 | 35.19 | 49.08 | 31.82 | 36.82 | 41.19 |
| 16 | TIES | 20% | 55.46 | 54.32 | 49.08 | 31.82 | 37.21 | 45.58 |
| | DARE | 20% | 53.69 | 58.67 | 49.08 | 31.82 | 37.08 | 46.07 |
| | **RMM** ($p = 2$) | 73% | 79.72 | 85.16 | 90.83 | 76.58 | 86.09 | 83.68 |
| | No merging | 100% | 92.31 | 85.86 | 94.15 | 86.38 | 88.62 | 89.46 |
| | Task-Arithmetic | 20% | 53.23 | 35.59 | 49.08 | 31.82 | 36.82 | 41.31 |
| 32 | TIES | 20% | 55.78 | 55.19 | 49.08 | 31.82 | 37.62 | 45.90 |
| | DARE | 20% | 55.89 | 33.74 | 49.20 | 31.82 | 36.85 | 41.50 |
| | **RMM** ($p = 2$) | 66% | 63.87 | 82.78 | 91.28 | 77.12 | 87.05 | 80.42 |
| | No merging | 100% | 92.51 | 86.03 | 94.38 | 87.18 | 90.15 | 90.05 |
| | Task-Arithmetic | 20% | 53.49 | 35.88 | 49.08 | 31.82 | 36.82 | 41.42 |
| 64 | TIES | 20% | 56.14 | 55.94 | 49.08 | 31.82 | 38.08 | 46.21 |
| | DARE | 20% | 53.51 | 33.68 | 49.08 | 31.82 | 36.82 | 40.98 |
| | **RMM** ($p = 2$) | 63% | 56.12 | 81.22 | 91.06 | 76.74 | 88.07 | 78.64 |
| | No merging | 100% | 92.68 | 86.14 | 94.50 | 87.33 | 91.05 | 90.34 |
| | Task-Arithmetic | 20% | 53.60 | 36.06 | 49.08 | 31.82 | 36.82 | 41.48 |
| 128 | TIES | 20% | 56.40 | 56.35 | 49.08 | 31.82 | 38.61 | 46.45 |
| | DARE | 20% | 50.52 | 34.96 | 49.08 | 31.82 | 36.92 | 40.66 |
| | **RMM** ($p = 2$) | 62% | 54.11 | 79.19 | 90.60 | 74.49 | 87.46 | 77.17 |

Table 6: Performance on GLUE benchmark for merging five RoBERTa-base models compressed with PT-SVD at various ranks ($r = 16, 32, 64, 128$).

## A.6  LLM USAGE

We used LLM only to improve the writing of this paper.

## A.7  QWEN2.5-3B RESULTS (YANG ET AL., 2024B)

| Method | Storage | QNLI | MRPC | SST-2 | MNLI | QQP | RTE | Average |
|--------|---------|------|------|-------|------|-----|-----|---------|
| No Merging | 100% | 61.56 | 81.57 | 96.44 | 90.58 | 90.77 | 77.98 | 83.15 |
| Task-Arithmetic | 17% | 50.54 | 70.61 | 54.13 | 35.33 | 64.07 | 50.90 | 54.26 |
| TIES | 17% | 50.49 | 72.06 | 54.01 | 35.85 | 64.06 | 50.90 | 54.56 |
| DARE | 17% | 50.50 | 70.32 | 54.01 | 35.25 | 64.05 | 50.90 | 54.17 |
| **RMM-2** | 56% | 56.25 | 72.23 | 94.95 | 90.48 | 88.48 | 55.23 | 76.27 |
| **RMM-3** | 76% | 59.89 | 74.78 | 96.10 | 90.61 | 90.20 | 61.73 | 78.89 |

Table 7: Performance on GLUE benchmark for merging six Qwen2.5-3B models compressed with LoRA at rank ($r = 32$).

