# OpenReview forum: "Towards Reversible Model Merging For Low-rank Weights"
_ICLR.cc/2026/Conference — Submitted to ICLR 2026_

### Official Review · Reviewer_Cgv9 · 2025-10-22

**Soundness:** 1
**Presentation:** 3
**Contribution:** 1
**Rating:** 2
**Confidence:** 3

**Summary:**

The paper relaxes the model merging problem into a compression–reconstruction setting. Based on this setting, this paper proposes a simple reversible model merging method that provides a closed-form solution to the optimization-based compression objective. The experiments demonstrate that the proposed method can maintain relatively good performance while significantly reducing the number of stored parameters.

**Strengths:**

1. The proposed method is simple and easy to implement.
2. The paper is clearly written and easy to follow.

**Weaknesses:**

1. This paper relaxes the model merging problem by assuming that the task index is known during inference. Compared with the conventional setting of model merging, this setting is overly simplified and reduces the practical applicability of the method.
2. The paper lacks evaluations on the latest models.
3. There is no comparison with model merging approaches based on LoRA, such as [1].
4. The experimental results appear underwhelming — with 69% of the parameters retained, the performance drops by more than 10%, which seems difficult to justify.

[1] Merging LoRAs like Playing LEGO: Pushing the Modularity of LoRA to Extremes Through Rank-Wise Clustering, ICLR 2025.

**Questions:**

1. A more appropriate baseline could be a comparison with quantized LoRA methods, which also aim to reduce storage requirements.

---

> ### Author Response · Authors · 2025-12-03
>
> We thank the reviewer for their comments and suggestions on the baselines.
>
> **Weakness 1: Assuming the task is known is oversimplified and reduces the practical applicability**
>
> The assumption of a known task index is common in nearly all model-merging works, as different tasks typically require distinct classification heads. Importantly, this is not a fundamental limitation of our method. If needed, one can train a small auxiliary router, e.g., a lightweight classifier that predicts the task index from the input text.
>
> **Weakness 2: Lack of evaluation on the latest models**
>
> Thank you for your suggestion. We have extended our experiments to include the Qwen2.5-3B model and provided the results for it in the Table 7 of the revised pdf.
>
> **Weakness 3: Lack of comparison with LoRA merging methods**
>
> Thank you for your suggestion. We have provided the KnOTS [1] merging baseline in the revised pdf and will add more LoRA based merging baselines as well as the LEGO [2] merging.
>
>
> **Weakness 4: Underwhelming results**
>
> The important point of our method is the fully controllable trade-off between performance and storage. Regarding the significant reduction of performance in low-rank weight spaces for previous merging baselines, our method enables superior performance. By increasing the p in RMM, the reconstruction quality improves, and the original model performance can be approached arbitrarily closely at the cost of additional storage. In other words, the 69% setting is simply one point on this curve rather than a fundamental limitation of the method. It is also worth mentioning again that this approach is completely data-free and one future work direction is to include some kind of calibration data and apply performance-aware reconstruction techniques to further improve the performance.
>
> **Question 1: Comparison with quantized LoRA methods**
> Thank you for your recommendation. We will add comparisons with quantized LoRA methods within the same storage budget as different RMM and rank settings to compare with another aspect of compression as well.
>
>
> [1] Model merging with SVD to tie the Knots
>
> [2] Merging LoRAs like Playing LEGO: Pushing the Modularity of LoRA to Extremes Through Rank-Wise Clustering

---

### Official Review · Reviewer_c6xK · 2025-10-27

**Soundness:** 2
**Presentation:** 2
**Contribution:** 2
**Rating:** 2
**Confidence:** 4

**Summary:**

This paper proposes Reversible Model Merging (RMM), a new method for combining multiple low-rank models (e.g., LoRA adapters). Instead of creating a single, often poorly performing, merged model, RMM learns a compact shared basis from which individual task-specific models can be accurately reconstructed on demand. This approach offers a tunable trade-off between storage and performance, and empirical results show it significantly outperforms existing merging techniques on compressed models.

**Strengths:**

* Effectively addresses the poor performance of standard merging techniques on low-rank models by reframing the problem as model reconstruction rather than aggregation.
* Demonstrates significant performance gains over established baselines across multiple model architectures (RoBERTa, ViT), tasks, and compression ranks.
* Offers an efficient solution for managing numerous task-specific models, with storage costs growing sublinearly with the number of tasks.

**Weaknesses:**

*   **Insufficient Discussion of Practical Overheads:** The primary trade-off of RMM is performance vs. storage, but another key factor is computational overhead at inference time. The paper does not quantify the latency introduced by the reconstruction step, which must be performed for each task. This could be a non-trivial cost, especially in latency-sensitive applications. Furthermore, the framework relies on an "oracle router" to select the correct task index `i`, and the practical feasibility and computational cost of this routing mechanism are not discussed.
*   **Lack of Connection to Intrinsic Dimensionality:** The core idea of RMM—finding a low-dimensional basis that can represent multiple task-specific updates—is closely related to the concept of intrinsic dimensionality in fine-tuning, e.g., [1]. Research has shown that the updates required for fine-tuning often lie in a very low-dimensional subspace. A discussion of how RMM relates to this concept and potentially a comparison to methods that explicitly estimate and use this intrinsic dimensionality could provide deeper insight into why RMM is effective and place it in a broader theoretical context.
*   **Limited Baselines for "Separate Merging":** The paper argues against "Combined merging" on the grounds of storage inefficiency and proceeds to compare RMM only against baselines using the "Separate merging" strategy. While the efficiency argument is valid, the empirical case for RMM would be stronger if it included a performance comparison against a "Combined merging" baseline (even if just for a smaller model or layer). This would help disentangle how much of the baselines' failure is due to the inherent flaws of their merging logic versus the limitations imposed by the separate merging of low-rank factors.
*   **Clarity and Precision of Mathematical Formulation:** While the overall method is clear, the notation in Section 5 could be more precise. The formulation for RMM is presented generically using `x_i` and `X` without explicit indices for the layer or the specific row/column being processed. Since the algorithm is applied independently to each row of matrix `A` and each column of matrix `B` for every layer, incorporating this into the notation would make the description less abstract and easier to map directly to the implementation.

[1] Intrinsic Dimensionality Explains the Effectiveness of Language Model Fine-Tuning

**Questions:**

See weakness

---

> ### Author Response · Authors · 2025-12-03
>
> We appreciate the reviewer’s recognition of the efficiency of our work and their constructive comments and suggestions on providing more baselines and adding discussions on other aspects of our work.
>
> **Weakness 1: Insufficient Discussion of Practical Overhead**
>
> Since the stored bases and coefficients for each task vector position are independent of each other, the reconstruction can benefit from parallel matrix multiplication using GPUs and the latency can be considered negligible. We also clarify that, if sufficient GPU memory is available, multiple task-specific models can be reconstructed in parallel in a distributed setup, removing the need to reconfigure the model for each new task input and further reducing overhead in practical deployments and therefore having the same GPU consumption as other merging methods with no additional overhead. We added this discussion in the revised pdf in Appendix A.5 as well.
>
> As for the router, almost every model merging paper assumes the task index is known since for classification, we have to change the classification head per task. However, there are works that also discuss the routing mechanism which could be just training a lightweight classifier that maps input text to the task index.
>
> **Weakness 2: Lack of Connection to Intrinsic Dimensionality**
>
> Thank you for your suggestion. We discussed this in the revised pdf in the introduction.
>
> **Weakness 3: Limited baselines**
>
> Thank you for your suggestion. We will add ablation study and comparisons on the combined merging, .i.e., first multiplying A and B matrices and then applying the merging algorithm as well.
>
> **Weakness 4: Mathematical notation**
>
> For brevity of notations, we had to represent task vectors as $x$ and indicate that this operation is done independently in both text and algorithm.

---

### Official Review · Reviewer_PQZt · 2025-10-30

**Soundness:** 3
**Presentation:** 3
**Contribution:** 2
**Rating:** 2
**Confidence:** 3

**Summary:**

This paper introduces Reversible Model Merging (RMM), a model merging method which creates a compact shared basis where individual merged models can be reconstructed. The experiments show that RMM outperforms several merging methods while being more storage efficient.

**Strengths:**

1. Novel perspective of relaxing the merging problem to allow maintaining more than just a single merged model.
2. Diverse task suites and model types in the experiments.

**Weaknesses:**

1. Limited comparison. Many modern SVD-based merging methods show good performance merging LoRA checkpoints (e.g. KnOTS [1], ISO [2], DRM [3]), as well as other non-SVD methods (e.g. PEFT’s cat [4]), though they aren’t compared or at least discussed.
2. Practical relevancy of merging low-rank compressed checkpoints is not clearly illustrated, reference to previous works supporting this is also lacking.
3. Natural baselines such as “combined” merging the checkpoints then compress, are not discussed or compared.

[1] https://arxiv.org/abs/2410.19735
[2] https://arxiv.org/abs/2502.04959v3
[3] https://arxiv.org/abs/2505.23117
[4] https://huggingface.co/blog/peft_merging

**Questions:**

1. I'm not quite convince regarding the relevancy or importance of “reversible.” If the individual checkpoints are already compressed, they should already be suitable for storage. Or if merged checkpoints need to be reversible, why merge them in the first place?
2. Usefulness of the storage reduction is debatable. As the merging is done post-training, the merged weight delta can simply be summed into the base weight, no extra parameters are to be kept.
3. Why are the tasks inconsistent between models? (QNLI, MRPC, SST-2, MNLI, QQP, RTE. CoLA, STS-B for RoBERTa vs. QNLI, MRPC, SST-2, MNLI, QQP,  BoolQ for OPT). What are the rationale for picking these subsets?


Writing
1. “an important limitation remains largely overlooked: even the most sophisticated merging strategies consistently fail to match the performance of the original individual finetuned models.” is not necessarily appropriate. As it’s not overlooked, but more of a goal the community is making progress toward.
2.  “This failure stems from two key factors: the limited expressive capacity inherent in low-rank representations, and misalignment of task-specific subspaces, each optimized independently, leading to severe interference when combined” needs further elaboration or citation.

---

> ### Author Response · Authors · 2025-12-03
>
> We appreciate the reviewer’s recognition of the novelty of our work and their constructive comments and suggestions on providing more baselines.
>
> **Weakness 1: Limited Comparison**
>
> Thank you for your suggestion. We have provided the results for the KnOTS [1] baseline method in the revised pdf and we will continue adding more baselines as well as discussing other methods. It is also worth mentioning that our method lies in data-free merging algorithms and thus cannot be compared with methods that require some held-out validation or calibration set.
>
> **Weakness 2: Practical relevancy of merging low-rank compressed models is not clearly illustrated**
>
> Thank you for pointing this out. We further elaborated on the importance of post-training low-rank approximation methods and discussed the benefit of them over LoRa in the introduction section of the revised pdf.
>
> **Weakness 3: Combined merging approach is not discussed or compared**
>
> We will add results for combined merging, i.e., first multiplying A and B matrices and then applying the merging algorithm in our ablation studies as well. However, we discussed why they cannot be practically relevant in section 4.
>
> **Question 1: Relevancy or importance of “reversible”**
>
> The argument behind the reversible merging paradigm is that the performance of individual models is always better than any merged model for the corresponding task, therefore, if we could efficiently and accurately go back to the individual model we would be able to achieve higher performance. Even though the individual models are already compressed, when the number of these compressed models grows there is a need to compress or merge them even further. The merged checkpoints do not have to be reversible, the reversible mechanism is our method. We merge them to save storage.
>
> **Question 2: Storage analysis**
>
> While it is true that the merged delta can be added to the base model, we are comparing the extra storage needed for our method within the size of low-rank deltas to give a fair and accurate comparison.
>
> **Question 3: Rationale for task selection**
>
> Since our approach’s storage efficiency relies on the number of models as well, we included experiments with different numbers of tasks and subsets (6 and 8 tasks for NLP and 7 tasks for vision). As for the task selection, we followed standard practice in the model merging literature and used the commonly adopted benchmark tasks.
>
> **Writing Concerns:**
>
> Thank you for your feedback. We rephrased that phrase and elaborated on the intuition behind the interference in the introduction section of the revised pdf.
>
> [1] Model merging with SVD to tie the Knots
>
> [2] Delta-CoMe: Training-Free Delta-Compression with Mixed-Precision for Large Language Models
>
> [3] Efficient Storage of Fine-Tuned Models via Low-Rank Approximation of Weight Residuals

---

### Official Review · Reviewer_YYBh · 2025-10-31

**Soundness:** 3
**Presentation:** 3
**Contribution:** 3
**Rating:** 4
**Confidence:** 3

**Summary:**

This paper proposes a new framework for merging multiple fine-tuned models that have been compressed using low-rank methods such as LoRA or post-training SVD. Traditional model merging techniques often fail when applied to low-rank models, leading to severe performance degradation due to limited representational capacity and task interference. To address this, the authors introduce Reversible Model Merging (RMM), which redefines merging as constructing a compact, reconstruction-capable model space rather than producing a single merged model. RMM learns a small set of shared basis components and task-specific coefficients through a closed-form SVD-based optimization, allowing each original model to be approximately reconstructed when needed. Experiments on multiple NLP and vision benchmarks show that RMM consistently outperforms existing merging methods like Task Arithmetic, TIES, and DARE, achieving much higher accuracy while maintaining efficient storage. The approach offers a tunable trade-off between performance and memory by adjusting the number of basis components, scaling efficiently to large multi-task or federated learning settings.

**Strengths:**

### Novel framing of the merging problem:
The paper redefines model merging as constructing a reversible and reconstruction-capable subspace rather than producing a single merged model. This is a clear conceptual shift from prior approaches that treat merging as a one-shot averaging process.

### Theoretical formulation with closed-form solution:
Unlike heuristic or sign-based methods (e.g., TIES, DARE), RMM provides a principled SVD-based framework that explicitly minimizes reconstruction error, offering mathematical interpretability and data-free optimization.

### 	Compatibility with low-rank compression:
The method operates directly on low-rank representations (LoRA, PT-SVD), addressing a practical gap where previous merging techniques fail due to limited expressivity and subspace misalignment.

### Flexible trade-off between performance and storage:
The introduction of the basis size p as a tunable hyperparameter enables a controllable balance between reconstruction fidelity and memory cost—an important design for scalable multi-task or federated learning scenarios.

**Weaknesses:**

### Lack of clarity in reconstruction generalization:
The paper does not clearly analyze whether the learned basis generalizes to unseen tasks or domains. RMM’s reversibility is validated only for known task models, which may restrict its applicability to dynamic or continual learning settings.

### Scalability validation remains limited in model scale:
While Section 6.3 demonstrates sublinear storage scaling across tasks, the evaluation is confined to relatively small models such as RoBERTa-base and OPT-1.3B. It remains unclear whether RMM maintains the same efficiency and reconstruction fidelity on larger-scale architectures. Validation on more recent and larger models (e.g., Qwen2.5-3B/7B or Qwen3-8B) would better demonstrate the robustness and scalability of the proposed method in realistic multi-billion-parameter settings.

### Lack of evaluation on modern LLM benchmarks:
The evaluation focuses primarily on classical NLP tasks (e.g., GLUE) and small-scale models. However, compared to recent works such as DARE, which include results on AlpacaEval, GSM8K, and HumanEval, this paper lacks validation on contemporary instruction-following or reasoning benchmarks that better reflect current LLM capabilities. Including such benchmarks would strengthen the empirical evidence for RMM’s effectiveness in modern large-scale settings.

**Questions:**

Tables 1 and 2 present the relative storage cost, but have you also measured the actual GPU memory consumption during inference or model reconstruction? Including such results could provide stronger evidence of RMM’s practical efficiency.

---

> ### Author Response · Authors · 2025-12-03
>
> We appreciate the reviewer’s recognition of the novelty and flexibility of our model merging paradigm and their constructive comments and suggestions on strengthening our experiments.
>
> **Weakness 1: Lack of clarity in reconstruction generalization**
>
> Our method reconstructs each individual model by expressing it within a merged basis. Therefore, the ability to generalize to unseen tasks fundamentally depends on the generalizability of the individual fine-tuned models themselves.
>
> **Weakness 2: Scalability validation remains limited in model scale**
>
> Thank you for your suggestion. We have extended our experiments to include the Qwen2.5-3B model and provided the results for it in the Table 7 of the revised pdf.
>
> **Weakness 3: Lack of evaluation on modern LLM benchmarks.**
>
> As for the reasoning, math or instruction-following benchmarks, since the number of models we can merge is relatively low (only 3 tasks in the DARE paper), it was hard to compare the baselines with our approach since our approach is more relevant for high number of tasks as we relax the number of models we can store. So, we decided to include the standard benchmarks for vision and NLP that are used by most model merging papers in the literature. However, we will keep exploring to add experiments for generative tasks as well.
>
> **Question 1: Lack of GPU Memory Usage Analysis**
>
> Thank you for this suggestion. For our method, there are multiple possible approaches for the reconstruction phase during inference (e.g., offloading to the CPU or leveraging fast, parallel matrix multiplications on the GPU). We also clarify that, if sufficient GPU memory is available, multiple task-specific models can be reconstructed in parallel in a distributed setup, removing the need to reconfigure the model for each new task input and further reducing overhead in practical deployments and therefore having the same GPU consumption as other merging methods. We have also discussed these points in the revised pdf in Appendix A.5.

---

### Meta-Review · Area_Chair_SSax · 2026-01-02

**Summary:**

All of the reviewers gave negative scores for this paper. Reviewer PQZt rightly brought up the questionable motivation (Q1) of the work. The authors remarked that the reversibility aspect of their proposed solution may be useful if the number of compressed models grow. However, this is in contradiction of their insistence that the work provides efficiency gains. Reviewer c6xK and Cgv9 brought up that there were limited baselines for separate merging. The authors responded inadequately. Overall, I agree with the reviewers that the paper needs more work.

**Reviewer Concerns:**

Reviewer c6xK and Cgv9 brought up that there were limited baselines for separate merging. The authors responded inadequately.

Reviewer PQZt rightly brought up the questionable motivation (Q1) of the work. The authors' response seems like a contradiction.

Overall, I do not think any score would have changed.

**Reviewer Scores:**

Overall, I do not think any score would have changed.

---

### Decision · Program_Chairs · 2026-01-26

Reject